# Cultural and Medicinal Use of Amphibians and Reptiles by Indigenous People in Punjab, Pakistan with Comments on Conservation Implications for Herpetofauna

**DOI:** 10.3390/ani12162062

**Published:** 2022-08-13

**Authors:** Saba Adil, Muhammad Altaf, Tanveer Hussain, Muhammad Umair, Jian Ni, Arshad Mehmood Abbasi, Rainer W. Bussmann, Sana Ashraf

**Affiliations:** 1College of Chemistry and Life Sciences, Zhejiang Normal University, Jinhua 321004, China; 2Department of Zoology, Sargodha Campus, University of Lahore, Sargodha 40100, Pakistan; 3Department of Forestry, Range and Wildlife Management, The Islamia University of Bahawalpur, Bahawalpur 63100, Pakistan; 4Department of Environment Sciences, Abbottabad Campus, COMSATS University Islamabad, Abbottabad 22060, Pakistan; 5Department of Ethnobotany, Institute of Botany and Bakuriani Alpine Botanical Garden, Ilia State University, Tbilisi 0105, Georgia

**Keywords:** folklore knowledge, conservation herpetology, Jhelum people, Pakistan

## Abstract

**Simple Summary:**

Humans have interacted with reptile and amphibian species for millennia. The current study was designed to collect knowledge about the use of amphibian and reptile species by the native peoples residing along the Jhelum and Chenab rivers in Punjab, Pakistan. To the best of our knowledge, this is the first quantitative assessment of the cultural uses of amphibian and reptile species in the study area. However, hunting, trade, and cultural use are the greatest threats to the diversity of the amphibians and reptiles in the studied area. These threats can potentially lead to their extinction. It is important to protect the highly endangered and vulnerable species employed in therapeutic medications, more specifically in terms of their conservation.

**Abstract:**

Amphibians and reptiles have interacted with humans for millennia. However, humans interact with amphibian and reptile species in different manners, which depend on their culture and traditions. This study was designed to better understand the interactions between amphibian and reptile species and their usage among the native peoples in the vicinity of the Jhelum and Chenab rivers, Pakistan. Information was collected through semi-structured interviews and questionnaires, and was analyzed by using different indices, including the frequency of citation, corrected fidelity level, fidelity level, relative importance level, and informant major ailment. Two amphibians and twenty-six reptile species were used in therapeutic medicine in the study area. Based on the cultural analysis, we found that *Naja naja* (black cobra) was highly cited across all cultural groups. A 100% *Fidelity Level* was calculated for the following species: *Naja naja* (eye infection), *Varanus bengalensis* (joint pain), *Eurylepis taeniolatus* (cataract), and *Acanthodactylus cantoris* (cancer). We found five endangered species in the study area, i.e., *Aspideretes gangeticus, A. hurum, Chitra indica, Varanus flavescens*, and *Geoclemys hamiltonii*, that were used to cure joint pain, muscle stretching and pain, backbone pain, paralysis, and psoriasis, respectively. Likewise, *Lissemys punctata andersoni*, a vulnerable species as labelled by the International Union for Conservation of Nature, was extensively used for the treatment of joint pain, body pain, paralysis, and arthritis in the study area. In terms of conservation, it is critical to protect the highly vulnerable and endangered species that are being used in therapeutic medicines. Our findings may be helpful for the conservation of amphibian and reptile species by helping to make an effective plan to prevent their extinction. The main threats to the diversity of amphibian and reptile species in the area are hunting, trading, and cultural use. These threats could potentially lead to the extinction of these species. Therefore, with the involvement of concerned authorities, e.g., local stakeholders, the Ministry of Climate Change, provincial wildlife departments, academia, and conservation managers, immediate conservation measures should be taken for the protection and sustainable utilization of medicinal species.

## 1. Introduction

Amphibians and reptiles are used for many purposes, including for food [1], art [2], pharmacology [3,4,5], calligraphy [6], culture [7], poetry [8], entertainment [9,10], religion [11], and clinical studies [12]. It has been observed that the interaction with amphibians and reptiles is valuable for maintaining good health [13,14]. The body products of amphibians and reptiles are utilized in ethnomedicine and nanomedicine [15,16,17]. Many species of amphibians and reptiles have significant value to humans [18,19]. Amphibians and reptiles are under direct threat due to the dangers of human activities such as road accidents [20], medicinal uses [17], illegal hunting, and trade [21], along with indirect threats such as the deforestation and modification of land [10,22,23,24,25,26,27,28]. Amphibians and reptiles are known as “diversified fauna” [29,30,31]. So far, an estimated 7850 amphibians and 10,450 reptiles have been documented around the world [32]. Khan [33,34] reported 24 amphibian species and 195 reptile species in Pakistan. Many different species of amphibians and reptiles are utilized in traditional and folklore medicine to cure ailments in other countries [24,31,35]. Humans exploit many of these species by using derived products such as meat, eggs, oil, blood, skin, shells, bones, and other body parts as natural products for tools, medications, decorations, food, and for magical and religious purposes [36,37].

In recent years, there has been increasing recognition for the importance of ethnobiological studies for biodiversity conservation. Ethnobiological research is essential for comprehending the sustainability of biocultural systems [38,39]. Moreover, ethnobiology offers critical insights into the customs of local peoples, enabling conservation efforts to collaborate with resource guardians to successfully promote the overall integrity of biocultural systems [40]. Cultural uses of animals (e.g., medicine, food, hunting, trade, entertainment, and religious practice) may promote beliefs and behaviors that aid in the conservation [41,42]. However, if these practices are being carried out in an unsustainable manner, or are influenced by economic, commercialization, and political factors, they may have a negative impact on or even endanger these animals [41]. It is important to consider other aspects, such as the changes in environment and climate, when analyzing how people use specific animal species for medicinal and cultural purposes [43,44]. The current challenge of biodiversity loss requires new strategies to be developed on a global scale [45].

Humans have interacted with animals for millennia, where the interaction reflects the impact from both culture and environmental conditions [42]. Depending on the desired usage and accompanying cultural characteristics, a single species can be employed in a variety of ways and for a variety of purposes by various communities [46,47,48]. Animals generally interact with humans due to their utility or the hazards they represent [47]. Furthermore, numerous myths, proverbs, and legends have arisen from these interactions and have been orally transmitted from generation to generation, affecting how the indigenous peoples interact with the animals [25,48]. Depending on the culture and region, different animals are exploited to different degrees. Direct exploitation is a major threat to biodiversity [49]. However, this includes timber exploitation and the acquisition of terrestrial as well as aquatic species for human use. Specifically, the use of reptiles as pets has been increasing around the world, generating high-value international trade with important implications for animal conservation. This trade of pets at high prices poses a hazard to numerous species that are often vulnerable [50,51]. Such species, including those recently described by the literature, frequently are of significant interest to collectors [52,53]. The presence of a species on Appendix I or II of CITES that would forbid or control its international trade is frequently compromised by a lack of data, economic interests, and the reality that conferences of the groups only occur every 3 years [54,55]. To ensure the survival of animal populations, conservationists must comprehend not only the ecological, but also the economic and cultural linkages that interconnect social and ecological systems into a single regional system, as well as the feedback that regulates these relationships.

Punjab is the second largest province in Pakistan after Baluchistan. The inhabitants of the Punjab province have diverse traditional knowledge and practices because of the great linguistic and cultural diversity present in the region [56]. The Jhelum people of Punjab widely use herptiles for ethnomedicinal purposes, e.g., *Aspideretes gangeticus, Daboia russelii, Ptyas mucosa*, etc. [7]. The diversity of herptiles has been documented by many authors [5,28,33,57,58,59], while the conservation aspects or direct uses have been observed only by a few researchers [3,7] in Pakistan. To the best of our knowledge, this is the first quantitative assessment of the cultural and medicinal uses of amphibian and reptile species in the study area. This study was designed to gain knowledge about the usage of amphibian and reptile species by the native peoples residing along the Jhelum and Chenab rivers in Punjab, Pakistan. We endeavored to give answers to the following questions: 1. How many amphibian and reptile species are employed in therapeutic medication in the healthcare system of Punjab, Pakistan? 2. Which species are the most frequently used? 3. What are the reasons for using amphibians and reptiles for medicinal purposes? 4. What are the key socioeconomic factors influencing the use of amphibians and reptiles for medicinal purposes (income, age, education level, and religion)? 5. What are the conservation consequences of using amphibian and reptile species for medicinal purposes?

## 2. Materials and Methods

### 2.1. Study Area and Native People

The present study was conducted between 2018 and 2020 in the Chenab and Jhelum riverine areas, i.e., Mandi Bahauddin (204 m elevation), Jhelum (234 m elevation), Gujranwala (231 m elevation), and Gujrat (233 m elevation) (Figure 1). The Jhelum people are agro-pastoralists, where they live in villages, grow crops, and use pastures for grazing livestock. Jhelum has no plains [60,61]. The languages of the local people are Punjabi and Pothohari [62]. The Chenab River originates in the state of Himachal Pradesh, India and continues into Pakistan [63]. While Punjabi is the common language, some people can speak Siraiki, Hindku, Pahari, and Urdu, while some people can also speak English to some extent [64,65,66]. The temperature starts near 0 °C in December and ends at 50 °C in June [64,65,66]. This agroforest land has a rich diversity of flora and fauna [23,33,67,68,69,70,71]. Most of the population is peri-urban, and it includes the Jutt, Sheikh, Arain, Gujjar, Malik, Mughal, Rana, and Butt casts.

### 2.2. Data Collection and Analysis

Before the start of the survey work, proper permission was obtained from the Department of Zoology at the University of Lahore, District Sargodha, Punjab. To acquire information on the therapeutic uses of amphibian and reptile species, semi-structured interviews and group discussions were conducted with 100 participants, after obtaining oral prior informed consent. Interviews were conducted during the daytime, and specimens (e.g., pictures, carcasses, etc.) were also collected during different visits. Informants were randomly gathered [41,72]. Some herpetofauna images were included in the questionnaires, and the semi-structured interviews contained questions about the local names of species and their ability to harm, as well as their applications in medicine, food, magic, narratives, superstitions, hunting, religion, and entertainment (Appendix A). Respondents’ age, sex, educational status, and linguistics were collected as demographic data. The questionnaires were first written in English (Appendix A), and then translated into Punjabi, Saraiki, and Urdu.

Amphibian and reptile species in the study area were directly identified by the respondents and confirmed. This was accomplished through photos included in the questionnaire and sent by e-mail or Facebook messaging. Amphibians and reptiles were confirmed by using *Amphibian and Reptiles of Pakistan* [33] for the correct identification and classification of the amphibians and reptiles in the study area [73]. The conservation status of each species was checked by consulting the Red List of Threatened Species of the International Union for Conservation of Nature and Natural Resources.

Informants were of a minimum age of 18 years and a maximum age of 91 years old (Appendix A). In the study’s survey, information was collected from male and female participants. Informants were told about the aims of the research after their permission to contribute to the data was obtained and they were guaranteed that their identities would be kept secret. The information about different usages of the animals’ body parts and their modes of application were shown in chord diagrams using the ‘circlize’ package in R software 3.6.1 [74]. Different indices were used to analyze the ethno-herpetological data, including the frequency of citation, corrected fidelity level, fidelity level, relative importance level, and informants of major ailment.

### 2.3. Frequency of Citation (FC)

The FC indicates the number of informants who reported the use of the animal species in medicine [75].

### 2.4. Fidelity Level (FL)

The FL was measured to determine the important species that were used by local peoples to treat specific ailments [75]. Its calculation is accomplished by using the following formula [76].
(1)FL (%)=(IMA/FC) × 100

IMA shows the number of informants who informed about the use of amphibian and reptile species for specific disease treatment, while FC is the total informers of a particular species.

### 2.5. Relative Importance Level (RIL)

The RIL was used to represent the level of popularity of different species in the study region. RIL is calculated using Equation (2), in which the number of respondents who claimed to use a single species (IMA) is divided by the sum of all respondents who claimed to use all species in the study area. The correction scale (CS) is used to distinguish between the popular and unpopular species. The RIL value ranges from 0 to 1.0. When animal species are used to obtain maximum ailment purposes, the RIL will increase from zero to the maximum value of ‘1’, while if the popularity of species for ailment purposes decreases, then the value will move from ‘1′ to ‘0’, showing divergence away from usage importance. [77,78].
(2)RIL=FC/FCt (0<RIL<1)

### 2.6. Corrected Fidelity Level (CFL)

The CFL was utilized as a factor of correction to find out the exact rank of animal species with dissimilar FL and RIL values. The CFL index was obtained by utilizing the following formula [77,78].
(3)CFL=FL×RIL

### 2.7. Ethics Approval

The proposed research on animals (especially amphibians and reptiles) was duly approved by the institutional committee of the Department of Zoology at the University of Lahore, Sargodha, Punjab, with a focus on the intellectual property rights of informants before the filed survey. In addition, the ethical guidelines and rules of the International Society of Ethnobiology (ISE) (http://www.ethnobiology.net/ accessed on 12 July 2018) were strictly followed.

## 3. Results

A total of two amphibians (7%) and twenty-six reptiles (93%) were used for therapeutic medicines in the study area. Data were documented from 100 respondents, whose ages ranged from 18 to 91 years (Figure 2). About 75% of the respondents were literate, with educations including Masters of Philosophy (2%), Masters degree (3%), Bachelor’s degree (2%), intermediate (10%), matriculated students (32%), middle school (20%), and primary school (6%). The use of amphibian and reptile species was more frequent among the illiterate people (25%). Most of the participants (75%) were from rural areas with an agriculture background (Figure 2). Most of the people in the study area were poor, and because traditional therapy is cheaper, they prefer folk medicine.

### 3.1. Principal Component Analysis

A principal component analysis was conducted with MD (medicinal), NR (narratives), SS (superstitions), TL (tool), CC (commercial use), ET (entertainment), FD (food), HF (harmful), MG (magic), EX (export), OR (ornamental), and REL (religious) values (Table 1). This analysis was used to highlight a significant difference in the use of amphibian and reptile species for cultural, food, and medicinal purposes, and was separated along the axis-1 (*p* < 0.05) as shown in Figure 3 and Figure 4. The significance of the PCA scores was confirmed by a one-way ANOVA, which calculated the analytic differences between the cultural and medicinal use of amphibian and reptile species. PC1 and PC2 elucidated 92.5% of the variance. The loadings of variables in PC1 showed that only *Naja naja* (black cobra) was positively correlated with cultural values while other species had positive correlations with medicinal use value (Figure 4).

### 3.2. Quantitative Assessment

#### 3.2.1. Fidelity Level (FL)

During the study, the Fidelity Level of amphibian and reptile species varied from 5.88% to 100% (Table 2). A 100% FL was noted for four species used to cure specific ailments, i.e., *Naja naja* for eye diseases and as an energy source to remove body weakness, *Varanus bengalensis* for the treatment of joint pain, *Eurylepis taeniolatus* for cataracts, and *Acanthodactylus cantoris* for cancer.

During the statistical analysis, only 6 species had RIL values of more than 0.70 (Figure 5). The highest value of RIL (1.00) was documented for *Bufo stomaticus* and *Hemidactylus flaviviridis* (Figure 5), followed by *Oligodon taeniolatus* (RIL = 0.83), *Lycodon aulicus* (RIL = 0.73), and *Naja oxiana* (RIL = 0.73) (Table 2). Only two species (*Bufo stomaticus* and *Hemidactylus flaviviridis*) were found to be more popular by the respondents, while other species were unpopular in the study area (Figure 5).

#### 3.2.2. Corrected Fidelity Level (CFL)

The highest CFL (36.5) was observed for *Psammophis leithii* for ailments involving snake, scorpion, wasp bite/sting, and eye infections, followed by *Uromastyx hardwickii* for treatment of joint pain (34.15). *Naja naja* and *Varanus flavescens* were cited for therapy of cancer (20.0) and paralysis (20.0), and *Varanus bengalensis* for joint pain (19.51) (Table 2).

## 4. Discussion

### 4.1. Socio-Demographic Data on Participants

Compared with the literate informers, whose exposure to modernization was higher, illiterate people in the research area were found to be less aware of the conservation aspects of reptile and amphibian species. Similar findings were found in Ethiopia [88,89] and Thailand [90]. Many inhabitants of the study area were heavily dependent on reptilian and amphibian species for a variety of purposes, including for supplementing their income. Selected informants belonged to different occupations, such as teachers, field workers, hunters, traditional healers, farmers, shopkeepers, house ladies, government employees, and laborers (Figure 2). We noted that rural informants had less knowledge about the conservation and sustainable use of species as compared with urban participants. Gathering socio-demographic data on participants (sex, age, educational level, occupation, and ethnicity) was thus particularly beneficial in social research, as this element plays a significant role in analyzing and interpreting the responses that were received [91].

### 4.2. Local Nomenclature

The local names of amphibians and reptiles were generally based on the fauna’s sound, environment, habitat, myth, morphological characteristics, and social links with *Homo sapiens*. As documented in Table 1, sap was utilized as a suffix synonym of eleven species of amphibians and reptiles (39.3%), such as *Bungarus caeruleus* (Sangchor sap), *Daboia russelii* (Kodian wala sap), *Lycodon aulicus* (Bhairia sap), *Naja naja* (Sheesh naag sap)*, N. oxiana* (Phaniar sap), *Oligodon taeniolatus* (gol dhari sap), *Psammophis schokari* (Saharai sap), *P. leithii* (Patta teer maar sap), *P. condanaru* (Siglee sap), *Ptyas mucosa* (Choh-mar sap), and *Ramphotyphlops braminus* (Dahga sap). Similarly, two species of lizard (7.14%) had the suffix “goh”, such as *Varanus bengalensis* and *V. flavescens*. Four species of lizard (14.3%) had the suffix “kirli”, i.e., *Laudakia agroransis* (Jungli Kirli), *Traplus agilis kistanensis* (Korh kirli), *Hemidactylus flaviviridis* (Gharailo kirli), and *H. flaviviridis* (Gharailo kirli). Local nomenclature of amphibians and reptiles is also based on their morphology; for example, *Lissemys punctata andersoni* has a green color known as “hara kachopra”, while *Acanthodactylus cantoris* has a blue tail known as “naili-push kirla”. Local names of documented amphibian and reptile species could also be linked with the habitats; for example, *Hemidactylus flaviviridis* being named “ghariallo kirli” and *Bufo stomaticus* being named “ghariallo daddo”, as both species live in the vicinity of houses (ghar). Sahrai sap was used as the name of *Psammophis schokari* because it lives in the desert area (sahara). The vernacular of one snake species was based on their morphology: *Ramphotyphlops braminus*, which has a thread-like structure known as dahga sap.

### 4.3. Cultural Uses of Amphibian and Reptile Species

As shown in Figure 3 and Figure 4, there were important dissimilarities in the use of amphibians and reptiles for cultural, food, and medicinal purposes, separated along the axis-1 (*p* < 0.05). We found that amphibian and reptile species were more commonly used for hunting, superstitious, and medicinal purposes (Figure 4). Reptiles have historically been a significant source of protein for humans across the world [43], and turtles are the most frequently exploited reptiles for human consumption, but snakes, lizards, and crocodiles can also serve as major food sources [92]. According to Alves and Souto [39], freshwater turtles have been reported as an important source of food for Amazonians during the dry season. In our study, non-Muslims ate five species of reptiles, such as *Aspideretes gangeticus, Naja naja**, Lissemys punctata andersoni, Chitra indica*, and *Aspideretes hurum* (Table 1).

Amphibians, on the other hand, are generally eaten in lesser quantities than reptiles. Frogs have been consumed locally in many countries as a high-protein source, including in Pakistan [72,93,94,95]. Amphibians have most likely always been eaten and utilized for cultural purposes in Gabon [96] and Cameroon [97]. Different active agents utilized as potential drugs have been isolated from amphibians [94,98], showing the medicinal importance of amphibians. According to Zhan et al. [94], 11 indole alkaloids and 118 bufadienolide monomers have been isolated from the *Bufo* spp., which exhibit a variety of in vitro and in vivo pharmacological activities, such as detoxification, anti-tumor, immunomodulation, and anti-inflammation.

Based on cultural applications, a cluster analysis revealed two groups of distinct species (Figure 6). The first cluster (G1) showed the highest cited species, including *Naja naja* (black cobra), which was highly cited across all cultural groups (Figure 6). The second cluster (G2) was made up of species with lower citations (Figure 6). Previous research has shown similar groupings. For example, Altaf [99] recorded two groups of wild animals from Punjab, Pakistan, used for cultural purposes by local residents, and eight primary clusters were documented by Rivera et al. [100] in the Castilla-La Mancha (Spain) mountains.

All documented amphibians and reptiles were exported from the Jhelum and Chenab rivers for food, medicine, and ornamental uses. Skins of snakes, such as *Oligodon taeniolatus, Naja naja, Lycodon aulicus, Psammophis leithii, Bungarus caeruleus, Psammophis condanaru, Daboia russelii*, *Naja oxiana*, and *Ptyas mucosa*, were used for decoration (Table 1). Ecologists noted that reptiles are intensively hunted in Pakistan for food, medicine, etc. [10,101]. According to the local informants, all species of amphibians and reptiles were regarded as poisonous and harmful.

### 4.4. Myths about Amphibians and Reptiles

Some common myths about amphibian and reptile species were also documented during the field survey (Table 1), as mentioned below. Similar myths were also noted by Altaf et al. [7].If someone kills a yellow-bellied common house gecko, God will give a reward to this person.If someone kills a Bengal monitor lizard or a yellow monitor lizard, the killer may die.All species have poison in their bodies, but they cannot bite because God has prohibited these species.All snakes are poisonous.All species of lizards have poison in their tails.If one of the partners in a pair of female or male snakes is killed by a human, the other will undoubtedly take revenge on the murderer.*Naja naja* and *Naja oxiana* change into human beings after the age of 100 years.Most people believe that the “Mankana”, a bone in a snake’s head, can absorb venom from any snake that bites a human.

### 4.5. Medicinal Uses of Amphibians and Reptiles

As documented in Table 2, 28 amphibian and reptile species were utilized to cure different ailments, i.e., wounds, anti-venom, asthma, backbone pain, cancer, cataract, body weakness, eye diseases, hepatitis C, allergy, joint pains, muscle stretching and pain, muscular weakness, paralysis, permanent flu, psoriasis, rheumatism, snake bite, scorpion bite, wasp bite, tuberculosis, tumor, underarm diseases, and vertebrae pain (Figure 7).

Higher values of indices can be linked to the fact that certain amphibian and reptile species were the most used species by the highest number of informants (Table 2). High fidelity level values confirmed that these amphibian and reptile species were more frequently used for the treatment of various ailments [102,103]. These results are supported by other scientists who reported amphibian and reptile species with high FL values that were employed to heal different human ailments [7,104], indicating that the native people of the study area held more information about the medicinal use of the documented species and less about conservation and sustainable information. Thus, the unfamiliarity of the people in the research region with respect to the ecology of amphibian and reptile species may cause their extinction.

*Varanus bengalensis*, a threatened species, was highly used in the study area for healing joint pain, body pain, paralysis, and arthritis (Figure 8). Hashmi et al. [105] reported that several tribes in Pakistan used the fat and oil of *V. bengalensis* as a salve for skin problems and to relieve rheumatic pain. The Bengal monitor lizard (*Varanus bengalensis*) is distributed throughout the Indus Valley, extending up to Las Bela in southern Baluchistan [106]. The lizard, when dipped in oil, is supposed to be used for the treatment of joint pain. Moreover, it is very popularly sold in Punjab, Pakistan (especially on footpaths, at bus stands, and at railway stations). The fat present in the body of a lizard is extracted and boiled down to the oil. The extracted oil is directly massaged on and around the treatment area. The active ingredients in oil are absorbed through the skin and into the body. There has been no medical research on lizard oil, but there have been clinical studies on a few of the popular herbs present in this oil mixture.

### 4.6. Body Part(s) Used

Oil was the most commonly used body part and was utilized in the synthesis of 26 recipes to treat various diseases, followed by the meat, bone, fat, skin, whole body, venom, and shell, which were used in 12, 9, 9, 7, 6, 2, and 1 recipes, respectively (Figure 9). People often exploited most animals by using derived products, such as oil (fat) [107,108], eggs, blood, meat, shells, bones, and skin. Several other parts were also used as food, ornaments, drugs, and for magical and religious purposes [109,110,111].

The people of the Jhelum and Chenab rivers in Punjab use oil to treat paralysis, muscle stretching, body pain, muscle weakness, broken bone treatment, asthma, tuberculosis, and provide energy to remove body weakness. According to Hashmi et al. [105], oil extracted from the belly fat of reptile species was used to treat skin infection, joint pain, and as an aphrodisiac lubricant. The local people used fat to cure psoriasis, allergies, and as an energy source to remove body weakness. For example, fat and oil from *U. hardwickii* were considered to improve sexual potency and in the treatment of body pain, joint pain, and paralysis in the study area. This has been linked to the treatment of erectile dysfunction, rheumatism, backbone pain, body pain, arthritis, blindness, fever, colds, and memory enhancement [7].

Jhelum persons used meat to cure vertebral pain, backbone pain, wounds, snake bites, scorpion stings, wasp stings, cataracts, and eyesight problems (Figure 9). A few tribal people from Sindh, Pakistan, such as the Kohli, Bahri, Bheels, Jogis, and Thani, consume meat for medicinal purposes to relieve rheumatic pain [103]. Local people specifically hunted animals for meat. Meat of different animal species was utilized in different folk therapies, e.g., abscess, anemia, infertility, asthma, strength, bronchitis, memory, immune enhancer, epilepsy, menorrhagia, fever, flue, paralysis, skin diseases, wound healing, and sexual potency [20,79,81,83,85,112,113,114,115].

### 4.7. Zoonotic Diseases

Wild animals and plants are very important for indigenous peoples and local communities for their cultural [116], medicinal [70,117,118,119,120], and esthetic values, and also serve as bioindicators [99]. Diseases can be transferred from animals to humans due to interactions with wildlife [121,122,123], and thus people who have close contact with animals can be at risk of zoonotic diseases [124,125]. Zoonotic pathogens can be transmitted from animals to humans, and transmitted from humans to other humans through sexual contact, vectors, aerosols, infected droplets, and oral transmission [126,127]. Many zoonotic diseases are transferred from amphibians to humans, including salmonellosis, sparganosis, and germs causing nausea, vomiting, and diarrhea [128,129]. Likewise, various zoonotic ailments are transmitted from reptiles to humans such as mycobacteriosis, pentastomiasis, and gastroenteritis [128,130,131]. This study found that direct usage of amphibians and reptiles has an influence not only on species diversity, but also on human health due to the documented spread of various zoonotic diseases.

### 4.8. Conservational Aspects of the Encountered Species

The design and integration of biodiversity conservation plans requires understanding both the human–animal interactions and local use of natural resources [132]. In this context, documenting indigenous knowledge about animal-based remedies is extremely useful for the development of policies for sustainable use and restoration of natural resources [87]. Ethnobiological studies, in addition to providing information about traditional uses of fauna in any region, also cover the economic, traditional, and cultural value of animal species in human societies, and thus make a significant contribution to animal conservation efforts [43]. We found that 67.8% of the encountered species have so far not been evaluated with regard to their conservation status (NE), while 17.7% of species are currently listed as endangered (EN), 7.1% (*Bufo stomaticus* and *Sphaerotheca breviceps*) as least concern (LC), 3.6% (i.e., *Lissemys punctata andersoni*) as vulnerable (V), and 3.6% (i.e., *Varanus bengalensis*) are listed as near-threatened by the International Union of Conservation of Nature (Table 2 and Figure 10).

Surprisingly, most reptile and amphibian species (91%) in the region showed signs of danger, even though only 9% of them are currently categorized as endangered by the IUCN. The use of animal species for medicinal and other traditional uses is, however, not the sole threat to animal biodiversity in any location. Changes in temperature and other forms of interactions in an ecosystem all play a role in endangering animal populations and biodiversity [43,44]. Given the pressing need for answers to the present biodiversity loss catastrophe [45], particularly that of animal species, techniques that assess the problem in all of its complexity are required. For this reason, ethnozoology offers itself as a multidisciplinary approach that approaches the problem in a more comprehensive manner [40].

## 5. Conclusions

The current investigation discovered that indigenous peoples in the study area still use a wide variety of amphibian and reptile species in their healthcare systems. Traditional applications of different species were documented, to help conserve the traditional knowledge related to their use among the native people in the vicinity of the Jhelum and Chenab rivers in the Punjab province, Pakistan. A total of 26 reptiles and 2 amphibians were used in traditional medicine in the study area. Our results showed that the local people in the study area have access to broad traditional information due to of their connection with amphibian and reptile species. Some species, such as *Aspideretes gangeticus, A. hurum, Chitra indica, Varanus flavescens*, and *Geoclemys hamiltonii* were extensively used for the treatment of various ailments. Hunting, trade, and cultural use are the greatest threats to the diversity of amphibian and reptile species in the studied area, possibly leading to their final extinction. However, the present data will be useful for the assessment of the direct impact on the native fauna of the study area. With the involvement of local stakeholders, concerned authorities, academia, and conservation managers, immediate conservation measures should be taken for the protection and sustainable utilization of medicinal species.

## Figures and Tables

**Figure 1 animals-12-02062-f001:**
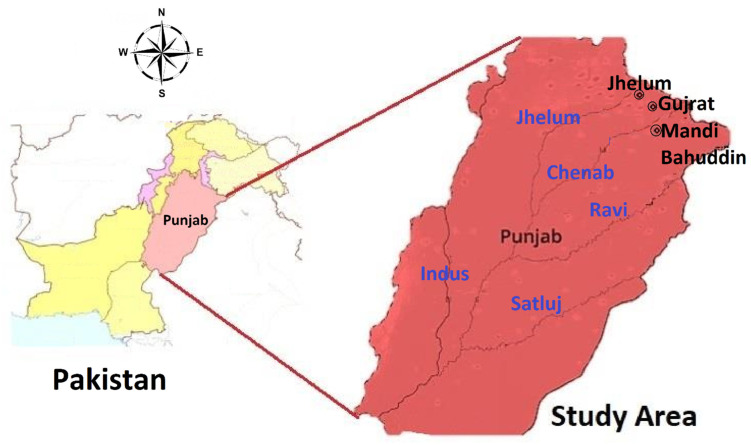
Map of the study area.

**Figure 2 animals-12-02062-f002:**
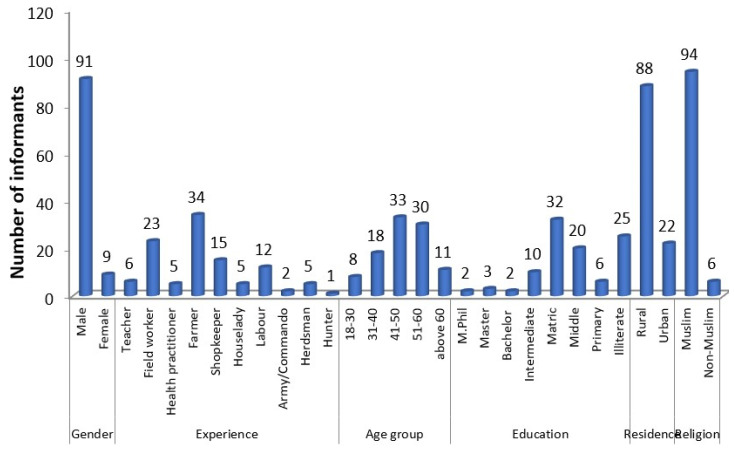
Numbers of study participants in the adjacent areas of the Chenab and Jhelum rivers, Punjab province, Pakistan. Respondents of different age groups, occupations, experiences, religions, residences, and educations were interviewed.

**Figure 3 animals-12-02062-f003:**
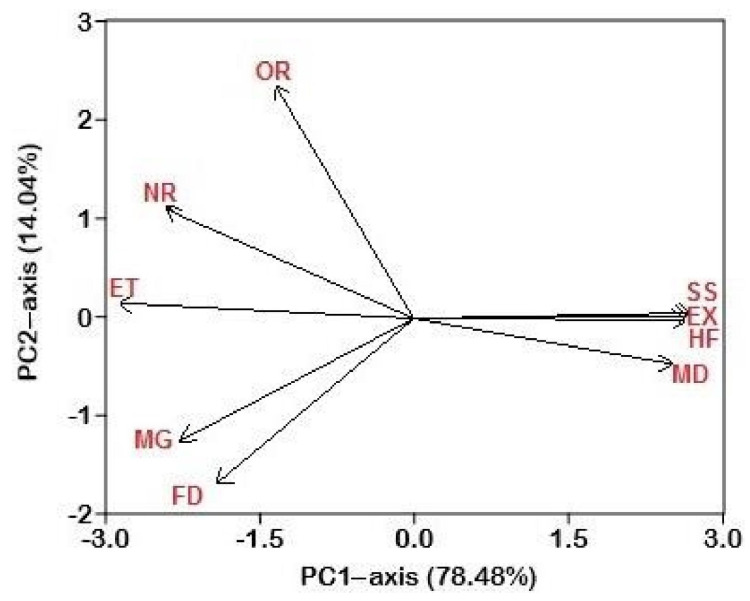
The principal component analysis (PCA) with the positions of the arrows relative to components 1 and 2, showing how strongly independent variables were correlated with each other. Plot of variables in the PCA conducted with MD (Medicinal), NR (Narratives), SS (Superstitions), CC (Commercial use), TL (Tool), FD (Food), HF (Harmful), MG (Magic), EX (Export), OR (Ornamental), ET (Entertainment), and REL (Religious).

**Figure 4 animals-12-02062-f004:**
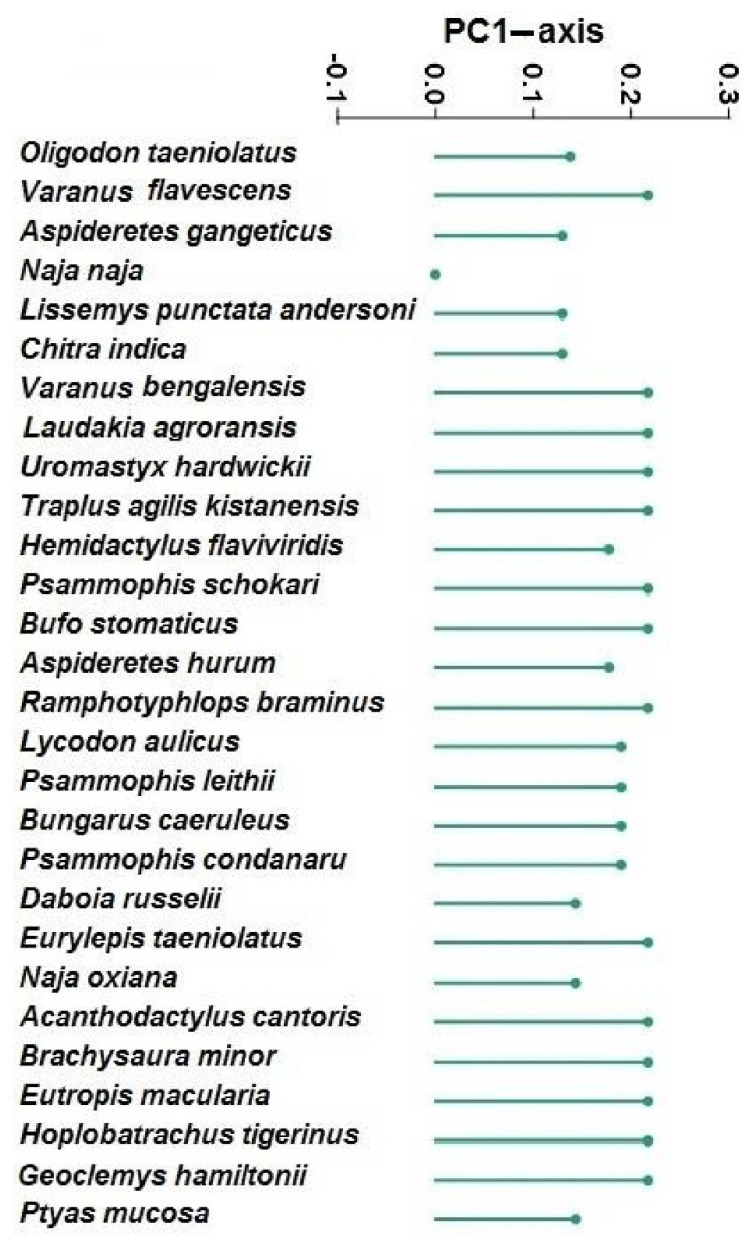
The loadings of PCA showing the correlation of different species with PC1 axis.

**Figure 5 animals-12-02062-f005:**
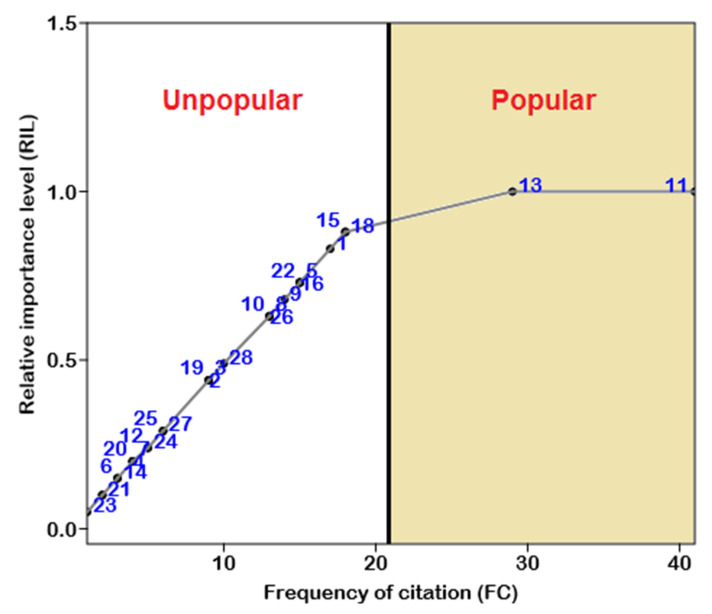
Relationship between the numbers of informants’ (FC) claimed use of certain species for particular diseases and relative importance level. The species’ relative importance level (RIL) was determined and classified as popular or unpopular. Numbers represent the species names as they appear in Table 2.

**Figure 6 animals-12-02062-f006:**
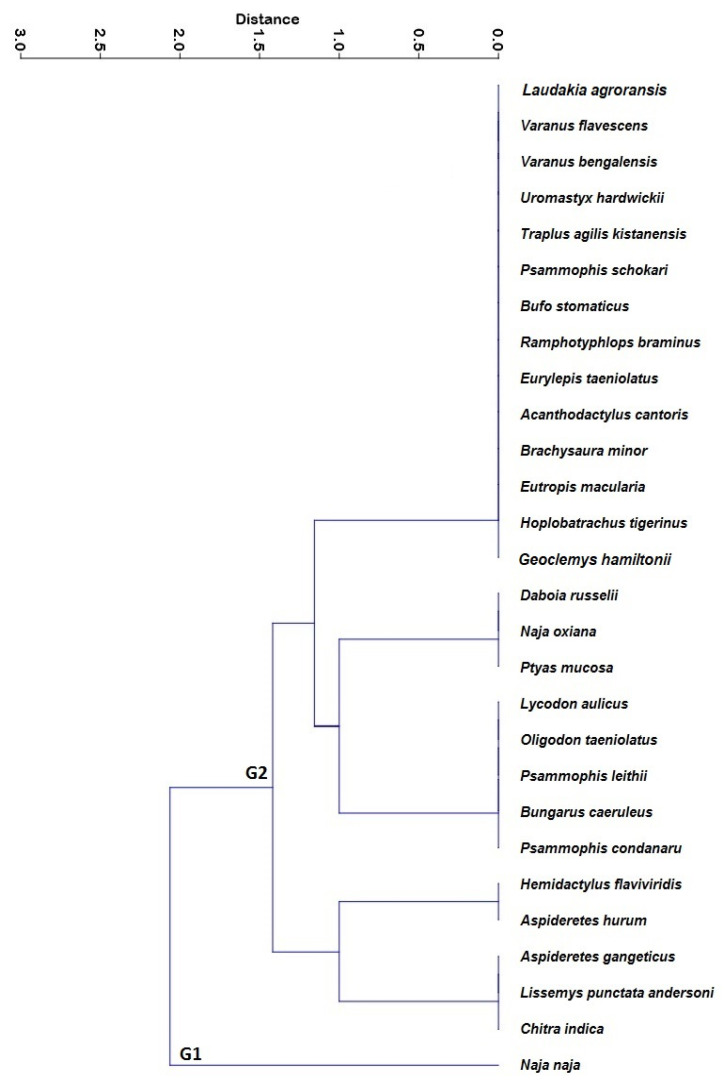
Cluster analysis showing the similarities between species with different variables (MD, NR, SS, CC, TL, ET, FD, HF, MG, EX, OR, REL) within the study area (G1, i.e., Group One and G2, i.e., Group Two).

**Figure 7 animals-12-02062-f007:**
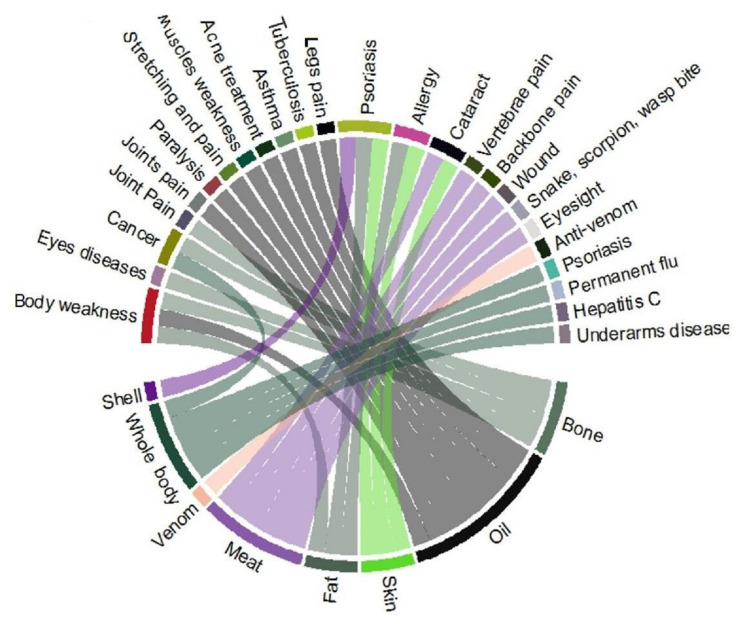
Body parts of animal species used in different recipes to treat various types of ailments.

**Figure 8 animals-12-02062-f008:**
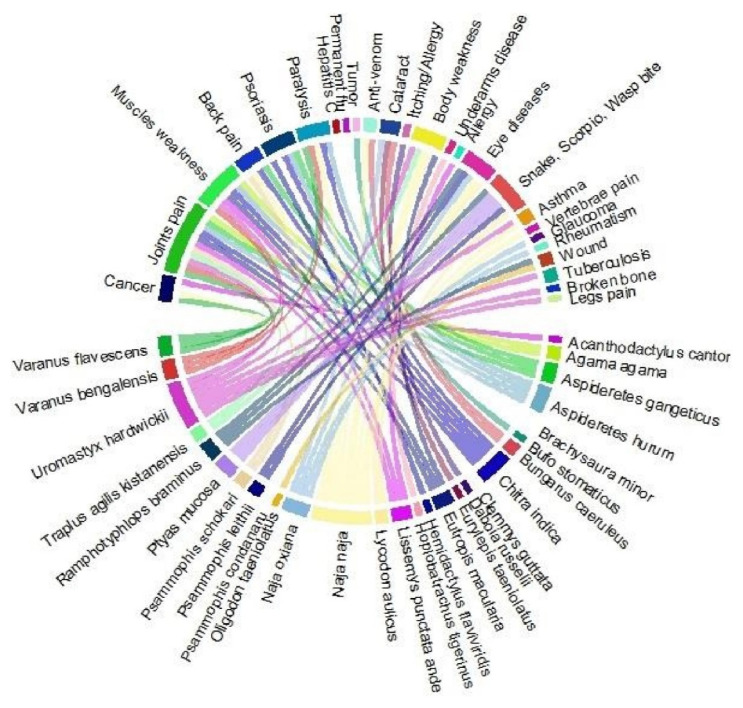
Animal species distribution according to the treatment of various ailments in Punjab, Pakistan.

**Figure 9 animals-12-02062-f009:**
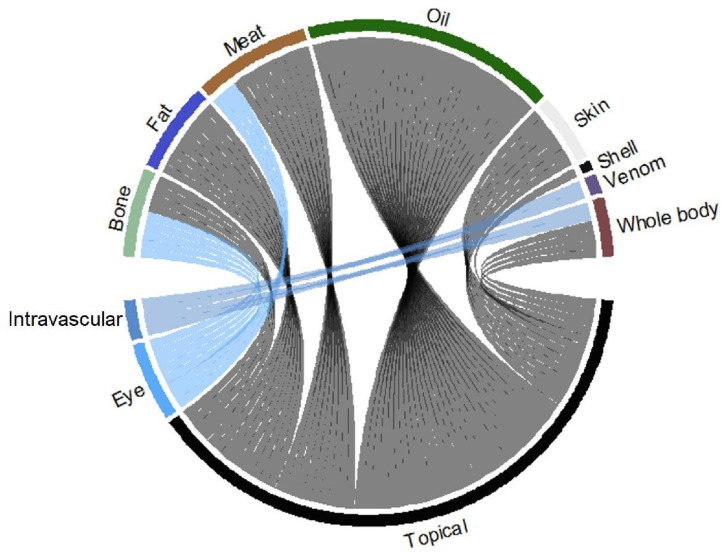
Relationship between different methods of preparation and administration of herbal remedies.

**Figure 10 animals-12-02062-f010:**
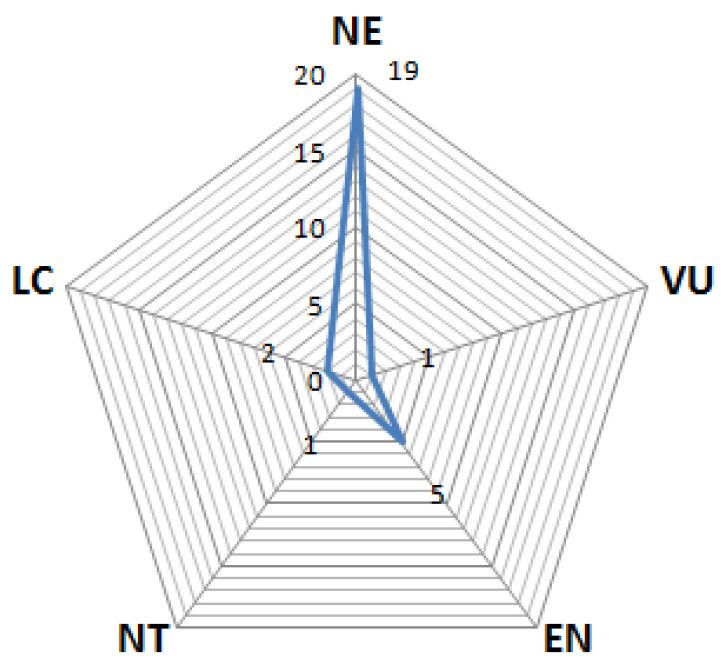
The conservation status of the species in the study area.

**Table 1 animals-12-02062-t001:** Ethnozoological data of herpetofauna.

Scientific Name Common Name Punjabi Name	Status	MD	NR	SS	ET	FD	HF	MG	EX	OR
*Oligodon taeniolatus* (Jerdon, 1853) Streaked kukri snake Gol dhari sap	NE	√	X	√	X	X	√	X	√	√
*Varanus flavescens* (Hardwicke & Gray, 1827) Yellow monitor lizard Goh	NT	√	X	√	X	X	√	X	√	X
*Aspideretes gangeticus* (Cuvier, 1825) Indian softshell Plaither	EN	√	X	√	X	√	√	√	√	X
*Naja naja* (Linnaeus, 1768) Black cobra Sheesh naag sap	NE	√	√	√	√	√	√	√	√	√
*Lissemys punctata andersoni* (Webb, 1980) Indian flap-shelled turtle Hara kachopra	VU	√	X	√	X	√	√	√	√	X
*Chitra indica* (Gray, 1830) Indian narrow-headed softshell turtle Karkuma	EN	√	X	√	X	√	√	√	√	X
*Varanus bengalensis* (Daudin, 1802) Bengal monitor lizard Goh	EN	√	X	√	X	X	√	X	√	X
*Laudakia agroransis* (Stoliczka, 1872) Agror wali agama Jungli kirli	NE	√	X	√	X	X	√	X	√	X
*Uromastyx hardwickii* (Strauch, 1863) Indus-valley spiny-tail ground lizard Sanda	NE	√	X	√	X	X	√	X	√	X
*Traplus agilis kistanensis* (Rastegar-Pouyani, 1999) Brilliant ground agama Korh kirli	NE	√	X	√	X	X	√	X	√	X
*Hemidactylus flaviviridis* (Ruppell, 1835) Yellow-bellied common house gecko Gharailo kirli	NE	√	X	√	X	√	√	X	√	X
*Psammophis schokari* (Forskail, 1775) Saharo-sindian ribbon snake Saharai sap	NE	√	X	√	X	X	√	X	√	X
*Bufo stomaticus* (Lutkin, 1862) Indus Valley toad Ghariallo daddo	LC	√	X	√	X	X	√	X	√	X
*Aspideretes hurum* (Gray, 1831) Peacock softshell turtle Kachhokuma	EN	√	X	√	X	√	√	X	√	X
*Ramphotyphlops braminus* (Daudin, 1803) Barhminy blind snake Dahga sap	NE	√	X	√	X	X	√	X	√	X
*Lycodon aulicus* (Linnaeus, 1758) White-spotted wolf snake Bhairia sap	NE	√	X	√	X	X	√	X	√	√
*Psammophis leithii* (Gunther, 1869) Steppe ribbon snake Patta Teer maar sap	NE	√	X	√	X	X	√	X	√	√
*Bungarus caeruleus* (Schneider, 1801) Common krait Sangchor sap	NE	√	X	√	X	X	√	X	√	√
*Psammophis condanarus* (Merrem, 1820) Indo-Burmese snake Siglee sap	NE	√	X	√	X	X	√	X	√	√
*Daboia russelii* (Shaw and Nodder, 1797) Russell’s chain viper Kodian wala sap	NE	√	√	√	X	X	√	X	√	√
*Eurylepis taeniolatus* (Blyth, 1854) Common mole skink Siddar	NE	√	X	√	X	X	√	X	√	X
*Naja oxiana* (Eichwald, 1831) Brown cobra Phaniar sap	NE	√	√	√	X	X	√	X	√	√
*Acanthodactylus cantoris* (Linnaeus, 1758), Blue tailed sand lizard Naili-push kirla	NE	√	X	√	X	X	√	X	√	X
*Brachysaura minor* (Hardwicke and gray, 1827) Hardwicke’s short tail agama Panj kirla	NE	√	X	√	X	X	√	X	√	X
*Eutropis macularia* (Blyth, 1853) Bronze grass skink Sap siddar	NE	√	X	√	X	X	√	X	√	X
*Sphaerotheca breviceps* (Schneider, 1799) Indian burrowing frog Daddi	LC	√	X	√	X	X	√	X	√	X
*Geoclemys hamiltonii* (Gray, 1821) Yellow-spotted turtle Chitra kuma	EN	√	X	√	X	X	√	X	√	X
*Ptyas mucosa* (Linnaeus, 1758) Rat snake Choh- mar sap	NE	√	√	√	X	X	√	X	√	√

MD (Medicinal), NR (Narratives), SS (Superstitions), CC (Commercial use), TL (Tool), ET (Entertainment), FD (Food), HF (Harmful), MG (Magic), EX (Export), OR (Ornamental), REL (Religious), NE (Not Evaluated), EN (Endangered), LC (Least Concern), VU (Vulnerable), and NT (Near Threatened).

**Table 2 animals-12-02062-t002:** The medicinal uses and statistical analysis of the herpetofauna in Punjab, Pakistan.

Scientific Name and Common Name	PU = MA	Medicinal Use	Reported Use	References	SI	IMA	FC	FL	RIL	CFL
*Oligodon taeniolatus* (Jerdon, 1853) Streaked kukri snake	B = T	Wounds			0	1	17	5.88	0.83	4.88
*Varanus flavescens* (Hardwicke & Gray, 1827) Yellow monitor lizard	O = T	Joint pain			0	3	9	33.3	0.44	14.63
	O = T	Paralysis				4		44.4		19.51
	O = T	Muscle stretching and pain				3		33.3		14.63
*Aspideretes gangeticus* (Cuvier, 1825) Indian softshell	SH = T	Psoriasis	Sexual potency, skin diseases, piles	[7]	0	1	9	11.1	0.44	4.88
	S = T	Joint pain				1		11.1		4.88
	F = T	Backbone pain				1		11.1		4.88
	O = T	Paralysis				3		33.3		14.63
*Naja naja* (Linnaeus, 1768) Black cobra	B = E	Eye diseases	Eyesight, leprosy, arthritis, cancer, sexual weakness, sciatica, snakebite, muscular pain	[79,80,81,82]	0	4	4	100	0.2	19.51
	F = T	Asthma				2		50		9.76
	M = T	Vertebral pain				1		25		4.88
	M = T	Backbone pain				1		25		4.88
	F = T	Energy source to remove body weakness				3		75		15.00
	S = T	Cancer				4		100		20.00
	B = T	Energy source to remove body weakness				2		50		10.00
	V = I	Anti-venom				2		50		10.00
*Lissemys punctata andersoni* (Webb, 1980) Indian flap-shelled turtle	O = T	Muscles stretching and pain	Piles, arthritis, allergy, acne, asthma, cough, dermatitis, epilepsy, bronchitis, burns, diabetes, urinary obstruction, backbone pain, lung diseases, malaria fever, diarrhea, indigestion, rashes, wounds, tuberculosis, sexual dysfunction	[82,83,84,85]	0.33	2	15	13.3	0.73	9.76
	F = T	Allergy				1		6.67		4.87
	O = T	Joint pain				2		13.3		9.73
*Chitra indica* (Gray, 1830) Indian narrow-headed softshell turtle	O = T	Muscle stretching and pain			0	2	3	66.7	0.15	9.76
	O = T	Joint pain				2		66.7		10.00
	O = T	Paralysis				2		66.7		10.00
	F = T	Psoriasis				3		100		15.00
	S = T	Backbone pain				2		66.7		10.00
*Varanus bengalensis* (Daudin, 1802) Bengal monitor	S = T	Cancer			0	1	4	25	0.2	4.88
	O = T	Joint pain				4		100		20.00
	O = T	Paralysis				3		75		15.00
*Laudakia agroransis* (Stoliczka, 1872) Agror wali agama	O = T	Muscle weakness			0	2	13	15.4	0.63	9.76
	O = T	Joint pain				2		15.4		9.69
*Uromastyx hardwickii* (Strauch, 1863) Indus Valley spiny-tail ground lizard	O = T	Joint pain	Enhance sexual power, treat earache, backbone pain, joint pain, headache	[79,82]	0.29	7	14	50	0.68	34.15
	O = T	Broken bones				2		14.3		9.71
	O = T	Asthma				2		14.3		9.71
	O = T	Tuberculosis				1		7.14		4.86
	O = T	Energy source to remove body weakness				4		28.6		19.43
	O = T	Leg pain				3		21.4		14.57
	O = T	Muscle stretching and pain				3		21.4		14.57
*Traplus agilis kistanensis* (Rastegar-Pouyani, 1999) Brilliant ground agama	O = T	Joint pain			0	1	13	7.69	0.63	4.88
	F = T	Energy source to remove body weakness				1		7.69		4.85
*Hemidactylus flaviviridis* (Ruppell, 1835) Yellow-bellied common house gecko	W = T	Psoriasis			0	2	41	4.88	1	4.88
*Psammophis schokari* (Forskail, 1775) Saharo-sindian ribbon snake	B = T	Joint pain			0	2	5	40	0.24	9.76
	M = T	Backbone pain				2		40		9.60
*Bufo stomaticus* (Lutkin, 1862) Indus Valley toad	W = T	Tumors	Allergy, pneumonia, dermatitis, ripened abscess, wounds	[82,86,87]	0	1	29	3.45	1	3.45
*Aspideretes hurum* (Gray, 1831) Peacock softshell turtle	O = T	Paralysis			0	2	3	66.7	0.15	9.76
	O = T	Muscle stretching and pain				2		66.7		10.00
	S = T	Psoriasis				2		66.7		10.00
	F = T	Joint pain				2		66.7		10.00
*Ramphotyphlops braminus* (Daudin, 1803) Barhminy blind snake	M = T	Wounds			0	1	18	5.56	0.88	4.88
	M = T	Snake, Scorpion, Wasp bite				3		16.7		14.67
	B = E	Eye disease				1		5.56		4.89
*Lycodon aulicus* (Linnaeus, 1758) White-spotted wolf snake	M = T	Snake, Scorpion, Wasp bite/sting			0	2	15	13.3	0.73	9.76
	B = E	Eye disease				1		6.67		4.87
*Psammophis leithii* (Gunther, 1869) Steppe ribbon snake	B = E	Eye disease			0	1	2	50	0.1	36.50
	M = T	Snake, Scorpion, Wasp bite/sting				1		50		36.50
*Bungarus caeruleus* (Schneider, 1801) Common krait	V = I	Anti-venom			0	1	18	5.56	0.88	4.88
	B = E	Cataract				2		11.1		9.78
*Psammophis condanaru* (Merrem, 1820) Indo-Burmese Snake	M = T	Tuberculosis			0	2	9	22.2	0.44	9.76
*Daboia russelii* (Shaw and Nodder, 1797) Russell’s chain viper	S = T	Allergy	Urine problem, hemorrhoids	[7]	0	2	4	50	0.2	9.76
*Eurylepis taeniolatus* (Blyth, 1854) Common mole skink	S = T	Cataracts			0	2	2	100	0.1	9.76
*Naja oxiana* (Eichwald, 1831) Brown cobra	M = E	Cataracts			0	2	15	13.3	0.73	9.76
	O = T	Rheumatism				2		13.3		9.73
	M = E	Glaucoma				1		6.67		4.87
	M = E	Eyesight				2		13.3		9.73
*Acanthodactylus cantoris* (Linnaeus, 1758), Blue tailed sand lizard	B = C	Cancer			0	1	1	100	0.05	4.88
*Brachysaura minor* (Hardwicke and gray, 1827) Hardwicke’s short tail agama	W = I	Cancer			0	2	5	40	0.24	9.76
	W = T	Permanent flu				1		20		4.80
	W = I	Hepatitis C				1		20		4.80
*Eutropis macularia* (Blyth, 1853) Bronze grass skink	O = T	Muscular weakness			0	1	6	16.7	0.29	4.88
	O = T	Joint pain				1		16.7		4.83
	F = T	Energy source to remove body weakness				1		16.7		4.83
*Sphaerotheca Breviceps* (Schneider, 1799) Indian burrowing frog	W = T	Underarm disease			0	2	13	15.4	0.63	9.76
*Geoclemys hamiltonii* (Gray, 1821) Yellow-spotted turtle	F = T	Psoriasis			0	2	6	33.3	0.29	9.76
*Ptyas mucosa* (Linnaeus, 1758) Rat snake	M = T	Snake, scorpion, wasp bite	Eyesight, epilepsy	[7,17]	0	3	10	30	0.49	14.63

B (body), N (nail), O (oil), F (fat), S (shell), F (fat), Bo (bone), M (molted skin), V (venom), E (eye), I (injection), W (whole body), C (consumed), T (topical), PU (parts use), MA (mode of application), relative importance level (RIL).

## Data Availability

All the data are presented in the tables and figures in the article or as a Appendix A, and further inquiries can be directed to the corresponding authors.

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
