# Peer review of "Cultural and Medicinal Use of Amphibians and Reptiles by Indigenous People in Punjab, Pakistan with Comments on Conservation Implications for Herpetofauna"

_animals, 2022, doi:10.3390/ani12162062_

Round 1

Reviewer 1 Report

Discuss on the values of harvesting in USD

Data on harvest of herpetofauna

Author Response

Respeceted Reviewer,

Thank you very much for your comments and suggestions. The comments and suggestions are valuable and very helpful for revising and improving our manuscript. We have revised this article according to the given comments and have made changes as they directed us (marked red).

Hope you will consider our efforts.

Reviewer 2 Report

Review of: “Conservation of Herpeto-Fauna Used in the Healthcare System of Punjab, Pakistan” 

 General Comments

This is an interesting but somewhat flawed manuscript. The idea of examining the attitudes of local citizens towards culturally important species of amphibians and reptiles is a good one and that does form the central portion of this paper. However, the attempts by the authors to make this paper into a “study was planned to conserve the diversity of amphibian and reptile species” misrepresents the data collected and what can be learned from it. Thus, the main thrust of the paper needs to be reconsidered.

In addition to this issue, the authors have attempted to use (or invent?) a series of quantitative terms such as “fidelity level” and “Corrected fidelity level” that are either just much simpler terms (such as percent or proportion) or which do not add in any substantive way to the point of the paper. In other cases, I simply could not tell what the authors were trying to do with these measures. 

The ms is also rather riddled with issues with the language and some biological details that need correction. Some of these are marked below under detailed comments, but I did not try and edit the entire ms in detail, as the authors need to work with someone with a better command of English before resubmitting the paper.

Finally, the sample size for the study (100 respondents) is really quite small and limits the scope of the paper, especially when you break the data set down by gender, education, rural versus city, etc. I also have reservations that these samples really were random, 

Detailed Comments

Line 25 (Abstract): Clemmys guttata is a North American native, so how it is of cultural significance in Pakistan needs explanation.

Lines 40-41: Is the use of the word “Love” really appropriate here?

Line 43: Starting with the word “Though” implies two clauses here, but that is not provided.

Line 47: This statement (“Amphibian and reptiles are known as the most important fauna”) needs better support. What do you mean by “most important?” In what context?

Lines 55-63: Some of this section is valuable, but a long explanation of CITES is not needed

Lines 66-68: This statement (“To the best of our knowledge, quantitative assessment of the cultural uses of amphibian and reptile species as conservation tools have never been reported before in the study area”) should form the major point of the introduction and the data set. A much shorter, narrower paper that did this would be of interest to the readership.

Lines 90 onward: Here and elsewhere in the ms, many words are placed in quotes (“department of Zoology”). This is incorrect and should be changed throughout the ms.

Line 92: Please specify how the respondents were selected randomly? Was this a stratified random approach or exactly what was done? This is a critical issue that could affect the entire data set.

Lines 92-96: This description of the questionnaire was so vague that I could not tell what the subjects were actually asked.

Lines 110-112: I have no idea what this sentence means: “The scale for the measurement of “folklore” was produced on basis of basic thoughts and features that local people linked with animals especially about amphibians and reptiles.

Lines 115-130 and onward. Here the authors try to come up with a series of sophisticated names for the metrics they use to quantify their results. Have these been used in previous studies? If so, those references need to be provided. If these are original, they need better justification. In some cases, you appear to be taking simple measures and just calling them by a different name. So (for example), isn’t “Fidelity Level” just the percent? (Line 125). In other cases, I could not tell what the measure refers to. For example, I was quite unclear about what you mean by a useful and useless species.

This section needs a lot of work. Because these are the basis for the data analysis to follow, I cannot really understand their results until all of this is clarified.

Line 152. Honestly, a sample size of 100 is very limited for a study such as this, especially when you stratify by age, gender, and education level. Your Figure 2 shows this quite well, as 90% of your sample is composed of males, making this a most biased sample.  This limits your inferences to a very narrow study.

I stopped commenting at this point as the results cannot be properly interpreted until these issues are resolved.

Author Response

(The authors gave the same response as above.)

Reviewer 3 Report

I have investigated the use of amphibians and reptiles by people in Asia and South America myself and read your manuscript with great interest. I have provided several comments and corrections for improvement.

The title is too short and does not reflect the quality and details of your paper. Something like the following would be better: "The cultural and medical use of amphibians and reptiles by indigenous people in Punjab, Pakistan with comments on conservation implications of the herpetofauna."

Several sentences require more details and others have unclear statements.

Overall, I find this paper very well written with clear methods and clear analyses and presentation of the results.

Author Response

Respected Reviewer,

Thank you very much for your comments and suggestions. The comments and suggestions are valuable and very helpful for revising and improving our manuscript. We have revised this article according to the given comments and have made changes as you directed us (marked red).

Hope you will consider our efforts.

Round 2

Reviewer 2 Report

Overall Comments:

 This is a revised version of a paper I reviewed earlier this year. Although there have been some improvements, the current version is not suitable for publication. There are two major issues at hand:

 1)      While the quality of the writing has been improved marginally, the language is still not close to being of sufficient quality for publication. I went through the first few pages of the paper and made extensive edits using Track Changes in Word so that the authors and the editor can see examples of the issues identified. Please note that the absence of comments after the first couple of pages simply means I did not examine those in any detail, NOT that those pages are acceptable.

 I completely understand the issues involved in writing a scientific paper in a second language and admire the authors for doing this. However, the fact remains that the current version is simply not of the quality needed for publication in this (or any) journal.

 2)      In my first review, I was quite critical of the methods and, I am sorry to say, that my opinion here has not changed. In addition to details noted on the Track Changes and below, my overall assessment is that this study was not designed correctly or, at the least, is so poorly explained that I clearly do not understand what was done and how.

 The most critical example of what I mean has to do with how the respondents were selected. As noted below and on the ms, in one place the authors say the respondents were selected randomly, without stating how this was done. However, later on in the same section, you state that the sample was confined to highly knowledgeable people, not a random sample as previously stated.

 This needs to be clarified. If the sample was truly random, please specify how this was done. If it was not random, this needs to be stated clearly, so we can see what the limits to your inferences are.

            Given these issues, the paper is not acceptable for publication. You simply cannot make broad inferences using data that were collected in a way that violates the basic principles of sampling. If all of this can be clarified and the paper re-written in an acceptable manner, a shorter, more focused paper could result. However, that will take considerable work and effort by the authors to make that happen.

Author Response

Respected Reviewer,

Thank you very much for your comments and suggestions. The comments and suggestions are valuable and very helpful for revising and improving our manuscript. We have revised this article according to the given comments and have made changes as you directed us (marked red). 

Hope you will consider our efforts
